# The BLA Benchmark: Investigating Basic Language Abilities of Pre-Trained Multimodal Models

**Xinyi Chen**
IvI
University of Amsterdam
x.chen2@uva.nl

**Raquel Fernández**
ILLC
University of Amsterdam
raquel.fernandez@uva.nl

**Sandro Pezzelle**
ILLC
University of Amsterdam
s.pezzelle@uva.nl

## Abstract

Despite the impressive performance achieved by pre-trained language-and-vision models in downstream tasks, it remains an open question whether this reflects a proper understanding of image-text interaction. In this work, we explore to what extent they handle basic linguistic constructions—active-passive voice, coordination, and relative clauses—that even preschool children can typically master. We present BLA, a novel, automatically constructed benchmark to evaluate multimodal models on these Basic Language Abilities. We show that different types of Transformer-based systems, such as CLIP, ViLBERT, and BLIP2, generally struggle with BLA in a zero-shot setting, in line with previous findings. Our experiments, in particular, show that most of the tested models only marginally benefit when fine-tuned or prompted with construction-specific samples. Yet, the generative BLIP2 shows promising trends, especially in an in-context learning setting. This opens the door to using BLA not only as an evaluation benchmark but also to improve models' basic language abilities.

## 1 Introduction

Powered by the Transformer architecture, extensive pre-training, and task-specific fine-tuning, recent language and vision models (Lu et al., 2019; Tan and Bansal, 2019; Li et al., 2019; Chen et al., 2020; Li et al., 2020; Su et al., 2019; Radford et al., 2021) have achieved unprecedented performance in many downstream multimodal tasks. Despite the impressive results, it remains an open question whether, and to what extent, this improvement goes hand in hand with a genuine understanding of image, text, and their interaction. In particular, the ability of models to handle linguistic skills that are essential to understanding an event or situation has recently been questioned. Hendricks and Nematzadeh (2021), for example, showed that these models fail in scenarios that require understanding verbs and verb arguments; Parcalabescu et al.

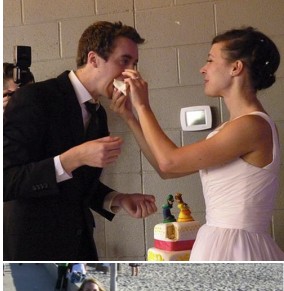
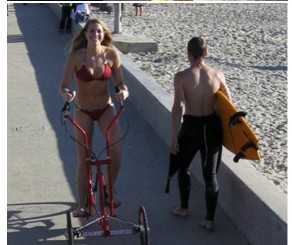
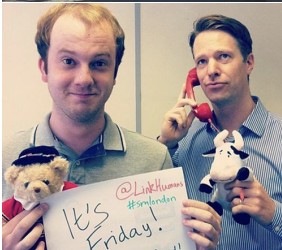

Active-Passive voice
**T:** the woman feeds the man.
**T:** the man is fed by the woman.
**F:** the man feeds the woman.
**F:** the woman is fed by the man.

Coordination
**T:** the man wears a wetsuit and carries a surfboard.
**T:** the woman wears a red bikini and rides a red bike.
**F:** the man wears a wetsuit and rides a red bike.
**F:** the woman carries a surfboard and wears a red bikini.

Relative Clause
**T:** the man who wears a gray polo holds a stuffed bear.
**T:** the man who wears a striped shirt holds a cow.
**F:** the man who wears a gray polo holds a cow.
**F:** the man who wears a striped shirt holds a stuffed bear.

Figure 1: One example for each of the linguistic constructions included in the benchmark. Sentences in each BLA dataset (both **T**rue and **F**alse ones) share the same template, with 3 slots being filled by as many arguments (NPs, predicates, or clauses), that we signal with different colors. E.g., in the active-passive construction, the bits in red and green are the subject and object of the sentence, respectively, while the blue bit is the predicate.

(2022) revealed a more generalized struggle of these models with phenomena that require grounding relations between objects; Pezzelle (2023) reported that their ability to ground language into vision is affected by the presence of semantically underspecified language, e.g., pronouns or locatives; Thrush et al. (2022) showed that no models appreciate the (substantial) difference between, e.g., *a lightbulb surrounding some plants* and *some plants surrounding a lightbulb*, pointing at flaws in

the way these models handle compositionality.

Arguably, all these previous studies require models to do more than plain language comprehension, including mathematical, spatial (Parcalabescu et al., 2022), pragmatic (Pezzelle, 2023), and compositional reasoning abilities (Thrush et al., 2022). While mastering these abilities is clearly desirable for any intelligent system, we notice that models may struggle with them due to their pre-training data and objectives. Indeed, these models are typically trained to verify whether a fragment of text is *about* the content of an image—via the Image-Text Matching (ITM) objective—which closely resembles the language comprehension tests administered to children to assess their lexical and grammatical abilities. To illustrate, in these tests, children are presented with a sentence, e.g., *The red monkey is being scratched by the blue monkey*, and asked to either pick the corresponding image from a set of alternatives (for an overview of this work, see Schmitt and Miller, 2010) or color the entities in a drawing accordingly (Pinto and Zuckerman, 2019). Consistent with their goal, these tests are aimed at excluding, or at least minimizing, the need for reasoning, which has been shown to be separate from language and to recruit different brain areas (Fedorenko and Varley, 2016).

In this work, we take inspiration from this line of research and investigate whether, and to what extent, language-and-vision models deal with language comprehension tasks that require virtually no reasoning abilities. We focus on three *basic* language constructions—active-passive voice, coordination, and relative clauses—that even preschool children typically master (Pinto and Zuckerman, 2019; Friedmann and Costa, 2010; Frizelle et al., 2017). We refer to these constructions as Basic Language Abilities (BLA) and propose an automatically constructed benchmark (see Figure 1) to assess pre-trained language-and-vision models, either in a zero-shot or in fine-tuning and in-context learning scenarios. We test several types of Transformer-based systems, i.e., CLIP, LXMERT, ViLBERT, BLIP2 and OpenFlamingo, and show that, while human (adult) speakers have no trouble at verifying these linguistic constructions, models generally struggle. Yet, the generative BLIP2 model shows promising trends, especially in an in-context learning setting. This reveals that, while BLA is a challenging benchmark, it can be used not only as an evaluation tool but also to improve SotA

models' basic language abilities—which are currently generally poor. We release the BLA benchmark and the code to reproduce our results at:
`https://github.com/shin-ee-chen/BLA`.

## 2 Related Work

### 2.1 Basic Language Comprehension Abilities

Language comprehension abilities—in children, but also in adults, e.g., L2 learners—are typically assessed in a multimodal setup: the subject is administered a sentence and some visual content and asked to verify whether the two match. Common paradigms are the Picture Selection Task (PST; Gerken and Shady, 1996), where the subject is given multiple candidate images to choose from, and the more recent Coloring Book (Pinto and Zuckerman, 2019), where the subject is presented with a single black-and-white drawing and asked to color objects in it according to the content of the sentence. Without any alternatives to choose from, in particular, the latter paradigm was introduced to minimize the recruitment of other executive functions connected to *reasoning*, such as selective attention and inhibition, that are not relevant to the assessment of genuine language comprehension.

Using these and similar paradigms, researchers demonstrated that linguistic constructions such as a sentence's active-passive voice, e.g., *The red monkey scratches/is being scratched by the blue monkey* (Pinto and Zuckerman, 2019), various types of coordination, e.g., *Grandma smiles and the girl sang* (Friedmann and Costa, 2010), and relative clauses, e.g., *He saw the girl that picked the flowers* (Frizelle et al., 2017), are generally mastered by preschool children across various languages, although with some differences due to the stimuli and, particularly, the experimental paradigm used. These results confirm that the comprehension of these linguistic constructions involves somewhat *basic* language abilities, as also indicated by previous evidence (Horgan, 1978; Diessel and Tomasello, 2001; McKee et al., 1998, *inter alia*).

In this work, we take inspiration from this line of research and aim at testing models for *basic* language comprehension abilities that require no or little reasoning skills. Our focused, controlled approach is novel compared to other work evaluating language-and-vision models, that we review below.

## 2.2 Language Abilities of Pre-Trained Multimodal Models

Motivated by the impressive performance of pre-trained multimodal Transformer models, a recent line of research investigated whether, and to what extent, this corresponds to a genuine understanding of visually-grounded language. Using FOIL (Shekhar et al., 2017a), a benchmark of minimally wrong image descriptions where all previous-generation models have proven to fail, some work (Hessel et al., 2021; Parcalabescu et al., 2022) showed that Transformer-based models can almost perfectly distinguish between correct and *foil* sentences in a zero-shot setting. This indicated that models are good at *grounding* nouns in vision, likely due to their ITM pre-training objective. Consistently, an intrinsic evaluation of the embeddings learned by these models showed that they are better at representing highly concrete—rather than abstract—words (Pezzelle et al., 2021).

Leveraging a FOIL-like paradigm, subsequent studies revealed that Transformer-based models struggle when dealing with verb arguments (SVO-Probes; Hendricks and Nematzadeh, 2021), negation (Dobreva and Keller, 2021), numbers, spatial relations (VALSE; Parcalabescu et al., 2022), and expressions requiring compositional abilities (WinoGround; Thrush et al., 2022; Diwan et al., 2022) or embedding semantically underspecified language (Pezzelle, 2023). Crucially, all this previous work focused on phenomena and tasks that require more than a *basic* language understanding to be properly mastered. As recently pointed out by Bugliarello et al. (2023), indeed, performing well on each of these benchmarks requires models to handle different skills, ranging from mathematics to pragmatics and reasoning abilities.

Inspired by the work discussed above, we take a novel perspective and assess language-and-vision models on their genuine lexical and grammatical competence. We consider three linguistic constructions—active-passive voice, coordination, and relative clauses—that have been shown to be mastered even by preschool children. In this paper, we refer to them as Basic Language Abilities.

## 3 The BLA Benchmark

In this section, we describe our Basic Language Abilities (BLA) benchmark.

### 3.1 Linguistic Constructions

BLA includes three types of linguistic constructions: active-passive voice, coordination, and relative clauses, which we briefly describe below.

**Active-Passive voice (AP)**  In active voice sentences, the agent of the action is the subject of the verb, as in *'the monkey scratches the mouse'*, while in passive voice sentences the form of the verb indicates that the subject is the receiver of the action; e.g., *'the mouse is being scratched by the monkey'*. Understanding the contrast between active and passive voice implies being able to verify whether two sentences with different syntactic structure may have the same meaning.

**Coordination (CO)**  Coordination, and in particular conjunction, binds together two properties that must hold. We focus on the coordination of verb phrases joined together via the conjunction *'and'*, e.g., *'the monkey eats an apple and smiles'*. Mastering this type of coordination implies being able to verify whether *both* predicates (*'eats an apple'* and *'smiles'*) apply to the subject of the sentence.

**Relative Clause (RC)**  Relative clauses are embedded clauses introduced by a relative pronoun that qualify an entity previously mentioned in the sentence. We focus on relative clauses attached to the subject of the sentence and introduced by the pronoun *'who'*, e.g., *'the monkey who scratches the mouse is tall'*. Understanding sentences with relative clauses implies identifying the entity qualified by the relative clause (e.g., *'the monkey who scratches the mouse'*) and verifying whether the predicate in the main clause applies (e.g., *'is tall'*).

### 3.2 Benchmark Format

We construct a dataset of natural images and template-based sentences for each of the linguistic constructions: active-passive voice (AP), coordination (CO), and relative clause (RC). Building on FOIL (Shekhar et al., 2017b) and FOIL-like paradigms, each datapoint in our benchmark consists of an image paired with 4 automatically generated sentences, two correct ones, hence *true*, and two incorrect ones, hence *false*. False sentences are identical to true ones with respect to their format and syntactic structure but contain a mistake that makes them semantically incorrect for the image. Concretely, in the false AP sentences the agent and the recipient are reversed; in the false CO sentences one of the conjuncts does not apply to the subject,

and in the false RC sentences the predicate of the main clause does not apply to the subject. Figure 1 shows one datapoint per linguistic construction.

## 3.3 Dataset Construction

To generate our datapoints, we use Visual Genome (Krishna et al., 2017), a dataset of natural images densely annotated with *objects* (bounding boxes around entities labeled with a WordNet synset; Miller, 1995), object *attributes* (such as color) and *relationships* between the objects in the image (predicates with their arguments).[1] The construction procedure includes the following steps:

**I. Selection of entities and predicates**  Firstly, we select images in Visual Genome that include at least two objects that are persons, which we verify using the WordNet synsets. For AP, we select images where the two persons are the arguments of the same transitive verb (one as the agent and the other one as the recipient).[2] We make sure that they belong to two different types (e.g., *man* and *woman*) or that they at least have two distinct attributes (e.g., *running player* and *sitting player*).

For CO and RC, we select images where the two persons are the subject arguments of two unique predicates, i.e., two different predicates per person that apply only to that person (e.g., *'wears a wetsuit'* and *'carries a surfboard'* for the man in Figure 1). Due to multiple annotations present in Visual Genome, checking whether two predicates are really different is not trivial. For example, a given person may be annotated as being the subject of two predicates, *'wears a t-shirt'* and *'wears a shirt'*. To capture the fact that these refer to the same relationship and should not be considered as different, we encode the predicates using Sentence-BERT (Reimers and Gurevych, 2019) and consider them independent predicates only if their cosine similarity is lower than 0.8. Moreover, for all datasets, we filter out the samples that involve reasoning language by means of handcrafted rules.[3]

**II. Minimum object size**  Secondly, we filter out images where the persons identified in the previous step, or the objects that are in a relationship with such persons, are too small. We consider them too

| | datapoints | vocab | sent. length | GRUEN |
|---|---|---|---|---|
| **AP** | 613 | 165 | 6.33 ±1.37 | 0.86 ±0.002 |
| **CO** | 654 | 827 | 10.46 ±1.44 | 0.84 ±0.003 |
| **RC** | 672 | 807 | 10.46 ±1.41 | 0.85 ±0.003 |

Table 1: Descriptive statistics of the BLA benchmark: number of datapoints, vocabulary size, average sentence length (number of tokens), and average GRUEN score. **AP**: Active-Passive voice. **CO**: Coordination. **RC**: Relative Clause.

small if their size is below a certain ratio between the area of their bounding box and that of the entire image. We use a threshold of 0.1% for persons and of 0.05% for other objects (such as *'bikini'* or *'stuffed bear'* in the examples in Figure 1).

**III. Sentence construction**  Thirdly, we construct true and false sentences using the templates in Table 5 and Table 6 in the Appendix by randomly filling them in with entities and predicates that meet the constraints above.[4] Since there may be a multitude of suitable entities and predicates per image, at this construction stage each image may end up being paired with more than one set of 4 sentences.

**IV. Grammar acceptability**  Finally, we encode each sentence with the GRUEN pre-trained language model (Zhu and Bhat, 2020) and obtain a score that, following previous work (Parcalabescu et al., 2022), we use as a proxy for grammar acceptability. We discard datapoints where any of the 4 sentences has a GRUEN score equal to or lower than 0.7. If, after this filter, an image is still paired with more than one set of 4 sentences, we only keep the one with the highest average GRUEN score.

## 3.4 Descriptive Statistics

In Table 1, we report the dataset size (number of datapoints per linguistic construction), vocabulary size, average sentence length, and average GRUEN scores. The AP dataset is the smallest of the three, the main reason being the limited number of transitive verbs (43) with arguments that meet our construction constraints. The sentences in CO and RC are constructed from the same set of entities and predicates. The slightly lower number of datapoints and vocabulary size in RC is due to more sentences with relative clauses being discarded by the grammar acceptability filter.

---

[1]More details on the annotation are in Appendix A.

[2]We check whether the predicates in the *relationships* field are part of the following list of transitive verbs: https://englishvocabs.com/transitive-verbs/184-transitive-verbs-list-in-english/.

[3]See Appendix B for details.

[4]We turn all verbs into 3rd person singular forms of the simple present tense using Python's *Pattern* library.

## 3.5 Human Performance

To assess the quality of our automatically constructed BLA benchmark, we run a human evaluation on 50 randomly selected datapoints for each linguistic construction dataset, using the Appen platform,[5] We create 4 HITs per datapoint, each displaying the image and one of the sentences. This results in a total of 200 HITs per dataset. The task is to judge whether the sentence is correct or incorrect given the image (therefore, random accuracy on this task is 50%). We ask 3 annotators (co-authors of the paper) to judge all HITs and compute human accuracy by considering the majority vote. Since human accuracy exceeds 85% on all three datasets, we conclude that, for human speakers, verifying the sentences included in BLA is a trivial task. In contrast, we verified that BLA cannot be solved by (even powerful) text-only language models, which do not fare better than chance in any of its tasks.[6]

## 4 Vision & Language Transformers

We experiment with 5 pre-trained language-and-vision models: three *discriminative* models (CLIP, ViLBERT, and LXMERT), and two *generative* models (BLIP2 and OpenFlamingo).[7] We briefly describe the models below.

## 4.1 Discriminative Models

**CLIP** CLIP (Radford et al., 2021) has two separate Transformer-based encoders, one for language, and one for vision (here, we use the ViT-B/32 version). These encoders are jointly trained on 400M <image, caption> pairs gathered from the web. It is trained through contrastive learning for predicting high similarity scores for paired <image, caption> and low scores for unpaired samples. We compute image-text alignment scores between an image and either its corresponding true or false sentences.

**ViLBERT** ViLBERT (Lu et al., 2019) is a dual-stream Transformer that encodes language and visual inputs separately before combining them via co-attention layers. It is pre-trained on the Conceptual Captions (Sharma et al., 2018) dataset with two learning objectives: multimodal masked learning (both word and object prediction), as well as

image-text matching (ITM). The pre-trained checkpoint we use is the one released by the VOLTA framework (Bugliarello et al., 2021).[8] Here, we use the pre-trained ITM head to straightly perform the binary classification (true/false) for each <image, sentence> pair in our benchmark.

**LXMERT** LXMERT (Tan and Bansal, 2019) is a dual-stream Transformer model that encodes vision and language via two separate streams and combines them via cross-model layers. The model checkpoint we use is pre-trained on the same exact data and with the same learning objectives as ViLBERT, again from the VOLTA framework.[9] Therefore, the two models are directly comparable.

## 4.2 Generative Models

**BLIP2** BLIP2 (Li et al., 2023) is a generative model that uses a Querying Transformer (Q-Former) to combine the information from a frozen large language model (LLM) and a frozen image encoder. The Q-Former contains two submodules—one image Transformer and one text Transformer—that share the same self-attention layers. The Q-former is trained in two steps: first, it connects the image Transformer submodule to a frozen image encoder to learn multimodal representations via image-text contrastive learning, image-grounded text generation, and image-text matching. Second, it performs vision-to-language generation by learning query embeddings that force the underlying LLM to generate text based on the visual information by the Q-former. We use BLIP2-FlanT5XXL.

**OpenFlamingo** OpenFlamingo (Awadalla et al., 2023) is an open-source reproduction of the Flamingo models (Alayrac et al., 2022). The model is pre-trained to generate text from a sequence of text tokens interleaved with images. It contains a frozen pre-trained CLIP-like image encoder and a frozen pre-trained large language model. The two components are connected via cross-attention layers that allow the language model to attend to features produced by the vision model. The models are pre-trained on the LAION-2B (Schuhmann et al., 2022) and Multimodal C4 (Zhu et al., 2023) datasets. In our experiments, we use CLIP ViT-L/14 as the vision encoder and one of the 3B ver-

---

[5] https://appen.com/

[6] Further details about our experiments with GPT-2 (Radford et al., 2019) are provided in Appendix H.

[7] We also experimented with FROMAGe (Koh et al., 2023) and MAGMA (Eichenberg et al., 2022) but decided not to include them in our study due to their limited ability to generate yes/no answers. More details are provided in Appendix F.

[8] Available at https://sid.erda.dk/share_redirect/aQCx8cLWK7.

[9] Available at https://sid.erda.dk/share_redirect/Dp1g16DIA5.

sions of the underlying language model.[10]

# 5 Exp 1: Zero-Shot Evaluation

To explore whether, and to what extent, the pre-trained models described in Section 4 can deal with linguistic constructions in BLA without any task-specific fine-tuning, we evaluate them on the three datasets in a zero-shot setting. For each dataset, we frame the problem as a binary task: given an <image, sentence> pair, the models are asked to evaluate whether the sentence is *true* or *false*.

ViLBERT and LXMERT can be straightforwardly evaluated on the binary task thanks to their pre-trained image-text matching (ITM) classification head. For CLIP, we compute similarity scores between the image and each sentence, and rank the four <image, sentence> pairs according to them. We consider the 2 top-ranked sentences as true and the 2 lower-ranked sentences as false.

We evaluate BLIP2 and OpenFlamingo by prompting. The prompt template used with BLIP2 is similar to the one proposed by Li et al. (2023) for Visual Question Answering: 'Question: Is the sentence [sentence] appropriate for this image? yes or no? Answer:'. For OpenFlamingo, following Awadalla et al. (2023), we use the following prompt template: '<image>Question: Is the sentence [sentence] appropriate for this image? yes or no? Short Answer:'. We let the models generate a response and consider 'yes' answers as true and 'no' answers as false.[11]

We use three metrics to measure model performance: (1) **accuracy**, measuring how well the models perform on the binary task, (2) **precision_true**, measuring how well models identify the true sentences, and (3) **precision_false**, measuring how well the models identify the false sentences.

## 5.1 Results

**All models lag far behind human performance** Results by all models are reported in Table 2. As can be seen, none of the models performs anywhere close to human performance on the BLA benchmark; indeed, most of them obtain results around chance level. While BLIP2 achieves a remarkably higher accuracy on AP (64%) than on the other two datasets, this result is still very far from 92%,

| Metric | Model / Humans | Task | | |
|---|---|---|---|---|
| | | AP | CO | RC |
| $Acc$ | ViLBERT | 50.57 | 49.81 | 49.96 |
| | LXMERT | 49.31 | 49.77 | 50.00 |
| | CLIP | 50.08 | 49.24 | 49.33 |
| | BLIP2 | **64.15** | **52.10** | **52.19** |
| | OpenFlamingo | 50.73 | 50.15 | 49.52 |
| | Humans | 92.00 | 90.00 | 85.00 |
| $P_t$ | ViLBERT | 50.43 | 49.80 | 49.97 |
| | LXMERT | 49.49 | 49.81 | 50.00 |
| | CLIP | 50.16 | 49.24 | 49.18 |
| | BLIP2 | **66.73** | **51.88** | **52.15** |
| | OpenFlamingo | 50.40 | 50.11 | 49.50 |
| | Humans | 90.00 | 79.00 | 79.00 |
| $P_f$ | ViLBERT | 50.84 | 49.82 | 49.96 |
| | LXMERT | 48.94 | 49.71 | 50.00 |
| | CLIP | 50.16 | 49.24 | 49.18 |
| | BLIP2 | **62.26** | **52.38** | **52.25** |
| | OpenFlamingo | 54.13 | 50.26 | 49.53 |
| | Humans | 94.00 | 95.00 | 95.00 |
| *chance* | | 50.00 | 50.00 | 50.00 |

Table 2: Zero-shot model performance and human performance on BLA. $Acc$: Accuracy. $P_t$: Precision_true. $P_f$: Precision_false. Scores are reported in percentage. The highest results for each metric and task are in **bold**.

i.e., human accuracy on this linguistic construction. Overall, this pattern of results reveals that, in a zero-shot setting, models struggle with the BLA benchmark, in line with previous work investigating other linguistic and reasoning phenomena (Thrush et al., 2022; Parcalabescu et al., 2022).

**BLIP2 is the best-performing model** The highest-performing model on the BLA benchmark is the generative model BLIP2. It outperforms OpenFlamingo and the best discriminative models with respect to all evaluation metrics (see Table 2). BLIP2 is the only model that consistently surpasses chance level on all three tasks—though by a small margin in both CO and RC—while the other models perform around or below chance level.

## 5.2 Discussion

The overall poor performance obtained in the zero-shot setting indicates that pre-trained multimodal models struggle with the language comprehension abilities evaluated by the BLA benchmark. This could be due to the way in which these models are typically pre-trained, i.e., maximizing cross-

---

[10]Concretely, as language model we use the instruction-finetuned model RedPajama-INCITE-Instruct-3B-v1.

[11]Both models always generated a *'yes'/'no'* answer.

modal alignment, which might not be fine-grained enough to account for the complex dynamics that intertwine language and vision. Performing well on BLA, indeed, requires understanding how entities interact with each other, how their attributes combine, and what attributes refer to which entity. Neither discriminative nor generative pre-trained models seem to handle these abilities.

At the same time, we notice that BLIP2 performs better than the other models, particularly on the active-passive construction. This advantage could result from the more varied data and pre-training objectives—image-text matching, image-text contrastive learning, and image-grounded text generation—of this model, which would help the model better understand verbs and verb arguments.

### 5.3 Error Analysis

We conduct an error analysis focused on the samples where the models consider all four sentences as either all true or all false. Considering that, in our dataset, each image is systematically paired with two true and two false sentences (see Figure 2), these cases are interesting since they indicate that models fail to make consistent predictions—intuitively, the sentences *the man holds the baby* and *the baby holds the man* cannot be true at the same time for a given image. This, in turn, would reveal that the models are unable to correctly identify the entities mentioned in the sentence.

For each model except CLIP,[12] we consider all the cases where all four sentences are assigned the same predicted label, either true or false. For BLIP2, these cases constitute 54.65%, 32.75%, and 38.55% of the samples in the AP, CO, and RC constructions, respectively. While these numbers may already seem quite high, we find out they are even higher in other models. For ViL-BERT, they increase particularly for AP (86.95%), with CO (49.43%) and RC (54.0%) experiencing a less dramatic increase. Similar percentages for AP, and a further increase for the other constructions, are observed for OpenFlamingo (88.5%, 68.5%, and 64.5% for AP, CO, and RC, respectively) and LXMERT (87.77%, 58.4, and 60.02%, resp.).

These patterns reveal that models are very often inconsistent with their predictions. This suggests they have a limited ability to identify the relevant entities, as well as their properties, in the image.

---

[12]Recall that, for CLIP, we use a ranking-based approach.

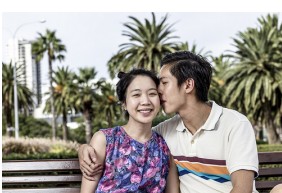
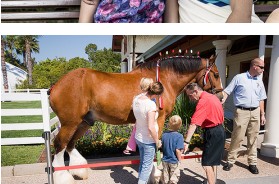

Active-Passive
**T:** the gentleman kisses the woman.
**T:** the woman is kissed by the gentleman.
**F:** the woman kisses the gentleman.
**F:** the gentleman is kissed by the woman.

Coordination
**T:** the man wears khaki pants and wears a blue shirt.
**T:** the woman wears a white shirt and wears jeans.
**F:** the man wears khaki pants and wears a white shirt.
**F:** the woman wears a blue shirt and wears jeans.

Figure 2: Two cherry-picked samples where all tested models predict that the four sentences are either all true (top) or all false (bottom), revealing an inconsistent behavior. **T** and **F** in the image refer to sentences that are **T**rue or **F**alse, respectively, in the BLA benchmark.

### 6 Exp 2: BLA-Specific Learning

To explore whether the models—which obtain poor performance in the zero-shot setting—can learn to deal with the linguistic constructions in BLA via some degree of task-specific learning, we expose them to a relatively little amount of data from each dataset. We use these samples to fine-tune the discriminative models and prompt the generative models to allow them to perform in-context learning.

In particular, we experiment with two types of BLA-specific learning, i.e., (1) we train and test a model with data from the same dataset (SD), and (2) we train a model with data from one dataset and test it with data from a different dataset (DD) in a cross-task setting. With the former, we evaluate whether the models can learn some key properties of the linguistic construction at hand by having exposure to the same phenomenon in other visual contexts. With the latter, we test whether the models can develop general language comprehension abilities that are key to all linguistic constructions.

We downsize each dataset to include 400 samples. Then, we randomly split it into training (100 samples), validation (100), and test (200) sets. While each sample in the validation and test sets has the standard format—one image and four corresponding sentences: two true sentences and two false ones—in the training set that we use to fine-tune or prompt our models we only keep two sentences per image. In particular, we randomly sample one true and one false sentence to keep the supervision of the models relatively little.

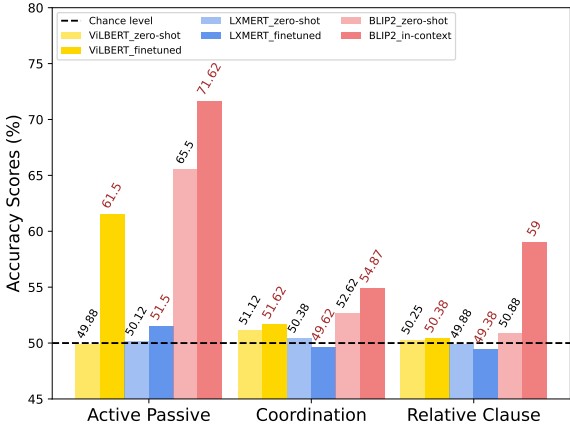

Figure 3: Comparison between model accuracies in zero-shot (lighter-color bars) and BLA-specific learning (darker bars). Results are obtained in the SD setting.

**Discriminative Models**   As CLIP does not have a classification head, we perform fine-tuning on ViLBERT and LXMERT. We use the training samples to fine-tune their ITM classification head. Further details are provided in Appendix D. In SD, models are fine-tuned, selected, and evaluated with data from the same dataset. In DD, they are evaluated with data from the test set of a different dataset.

**Generative Models**   We perform in-context learning with BLIP2 only. For OpenFlamingo, our preliminary investigations revealed that, with our data and task, using the in-context learning setup proposed in the original model paper led to nonsense outputs in many cases. The same was observed when using the same in-context learning setup that we used for BLIP2, which we describe below.[13]

In BLIP2, we perform in-context learning using samples from the test set of a given dataset. In SD, for each <sentence, image> pair that we aim to evaluate, we pick one true and one false sentence belonging to the same datapoint. Compared to the sentence under evaluation `[target]`, these sentences have either a different voice (AP) or describe attributes for a different person (CO and RC). In BLIP2, we fill these sentences into a template that we use as a prompt for the model. For example: 'Question: Is the sentence `[true]` appropriate for this image? yes or no? Answer: yes. Question: Is the sentence `[false]` appropriate for this image? yes or no? Answer: no. Question: Is the sentence `[target]` appropriate for this

---

[13]Further details about these preliminary experiments and corresponding prompting setups are reported in Appendix G.

| Metric | Model | Improvement ($\Delta P$) | | |
|--------|-------|------|------|------|
| | | AP | CO | RC |
| $P_t$ | ViLBERT | 11.00 | 0.75 | 1.00 |
| | LXMERT | 4.50 | 1.25 | -1.00 |
| | BLIP2 | -2.05 | 0.25 | 4.09 |
| $P_f$ | ViLBERT | 11.00 | 0.75 | 1.00 |
| | LXMERT | 4.50 | -1.69 | -0.31 |
| | BLIP2 | 18.74 | 42.34 | 46.50 |

Table 3: Precision improvement after BLA-specific learning (SD setting) compared to the zero-shot results. $P_t$ is precision_true, $P_f$ is precision_false.

image? yes or no? Answer:'.

In DD, the setup is the same as above, except that the `[true]` and `[false]` sentences are from another BLA dataset (different linguistic construction), and yet about the same image (same entities and attributes). While images in CO and RC greatly overlap, images in AP do not overlap much with those in the other two datasets. Therefore, we only experiment with CO and RC in this setting.

## 6.1 Results

**BLA-specific learning generally helps**   As reported in Figure 3, BLA-specific learning has a generally positive impact on model performance. The generative BLIP2 is shown to improve on all three datasets, which confirms the effectiveness of the in-learning setup in boosting this model's comprehension abilities. As for discriminative models, ViLBERT experiences the greatest accuracy improvement compared to the zero-shot setting on AP, and a smaller (though consistent) improvement on the other two tasks; LXMERT, in contrast, does not seem to equally benefit from fine-tuning, except perhaps for a little boost on the AP construction.

**BLIP2 is the best overall model**   As shown in Figure 3, BLIP2 is again the overall best model across the board. Indeed, it outperforms the best discriminative models by 10.1, 3.2, and 8.6 accuracy points on AP, CO, and RC, respectively. Moreover, it is the model achieving the higher relative improvement in accuracy on all BLA datasets over the zero-shot setting. It is worth mentioning that, looking at the relative improvement in *precision* obtained by various models over the zero-shot setting (Table 3), BLIP2 exhibits a fairly high improvement on all tasks, particularly CO and RC, with respect to precision_false. That is, task-specific learning particularly helps the model to better spot

| Model | Task for Learning | Improvement on Evaluation Tasks | | |
| --- | --- | --- | --- | --- |
| | | AP | CO | RC |
| ViLBERT | AP | - | -2.24 | **0.75** |
| | CO | -0.38 | - | -0.37 |
| | RC | 0 | -0.87 | - |
| LXMERT | AP | - | -1.38 | -0.38 |
| | CO | -0.12 | - | **0.5** |
| | RC | -0.24 | -0.38 | - |
| BLIP2 | CO | - | - | **11.25** |
| | RC | - | **6.94** | - |

Table 4: Accuracy improvement after BLA-specific learning (DD setting) compared to the zero-shot results

false sentences, which in turn allows it to achieve an overall higher performance on the tasks.

**Active-passive voice is the most easily learned** Overall, the active-passive voice construction appears to be the one that models can learn to a greater extent. While BLIP2 achieves an accuracy of more than 70%, ViLBERT ranks second with a respectable 61%, which is an even more notable result considering the performance around chance level in the zero-shot setting. LXMERT also outperforms the results obtained in zero-shot learning, though by a much smaller margin.

**Cross-task learning is only effective for BLIP2** As reported in Table 4, BLIP2 is the only model across the board that benefits from learning about a linguistic construction that is not the one under investigation (our DD setting). As can be seen, the improvement by BLIP2 on RC after having been exposed to CO exceeds 11% accuracy compared to the zero-shot learning, while the improvement in the other direction (RC to CO) is around 7%. This is not the case, instead, for the discriminative models, for which the role played by this setup is either insignificant or even detrimental.

## 6.2 Discussion

The results presented above generally show that BLA-specific learning has an overall positive role in helping models understand the linguistic constructions included in the benchmark. This suggests that having (even limited) experience with how these linguistic constructions work in visually-grounded contexts is beneficial for these models,

which are shown to improve their performance over the zero-shot setting. In particular, BLA-specific learning helps the generative BLIP2, which is shown to improve not only in the SD setting but also in DD, where examples of other linguistic constructions are provided. This pattern is encouraging and suggests that understanding these linguistic constructions may underlie some common basic language abilities dealing with the semantic properties of entities, attributes, and predicates, and their interaction with the image.

Yet, their performance on the benchmark is still far from human performance, with the best overall model (BLIP2) lagging 20 accuracy points behind human accuracy on AP, the highest-scoring dataset. At the same time, some linguistic constructions appear to be more challenging to learn than others, with coordination experiencing much lower improvement compared to AP and RC. On the other hand, AP stands out as the construction that can be best learned by the models, possibly due to the fact that it requires models to ground the entities—but not necessarily their attributes.

## 7 Conclusions

We introduced a novel benchmark, BLA, aimed at investigating how well multimodal models understand basic linguistic constructions—active-passive voice, coordination, and relative clauses. We showed that the linguistic constructions in BLA are challenging for current language-and-vision models, which lag well behind human performance. Yet, the recent generative model BLIP2 exhibits a better performance than discriminative models, both in the zero-shot and task-specific learning setting. We highlight that prompting generative models with examples embedding both the same or a different linguistic construction is a promising method to improve their understanding of specific linguistic constructions. This opens the door to using BLA not only to evaluate pre-trained models but also to improve their basic language abilities.

## Limitations

The BLA benchmark currently contains three tasks while more can be added for a more comprehensive understanding of basic language ability of multimodal models. But our pipeline can be easily adapted to construct more tasks. In the experiment, we only investigated a limited number of models, which include (i) generative models (BLIP2 and

OpenFlamingo), (ii) discriminative models with an image-text classification head (ViLBERT and LXMERT) and (iii) discriminative models with cross-modality similarity scores (CLIP), which we believe our selections are representative of current mainstream Vision-and-Language (V&L) models. Due to the in-context learning constraints for BLIP2, we only investigate its BLA-specific cross-task learning setup on Coordination and Relative Clause tasks. The two tasks are constructed with the same pipeline and contain similar descriptions of human attributes, so more investigation can be done to explore whether the model can improve with learning examples that are more semantically different.

## Acknowledgements

We are grateful to Iacer Calixto for the valuable feedback on a preliminary version of the dataset and experiments. We want to thank Emanuele Bugliarello for his help with the VOLTA framework and Hongyi Li for his support with the Open-Flamingo model evaluation. We also thank the members of the Dialogue Modelling Group at the ILLC and the members of the IRLab at IvI, University of Amsterdam, for their insightful feedback on a draft of the paper and the experiments. Xinyi Chen is funded by the project LESSEN with project number NWA.1389.20.183 of the research program NWA-ORC 2020/21, which is (partly) financed by the Dutch Research Council (NWO). RF is supported by the European Research Council (ERC) under the European Union's Horizon 2020 research and innovation programme (grant agreement No. 819455).

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

## A Visual Genome

Visual Genome (VG) is a dataset and a knowledge base that annotates 108K images with each image having an average of 35 objects, 26 attributes, and 21 pairwise relationships between objects. It contains seven main components —*region descriptions*, *objects*, *attributes*, *relationships*, *region graphs*, *scene graphs*, and *question-answer pairs*—together with information for *image*. We only used *objects*, *attributes* and *relationships* for our dataset construction and extracted the image size information from *image*. One example of the VG dataset and its corresponding annotations can be found in Figure 4.

## B Avoiding Reasoning in BLA Construction

By analyzing the VG annotations that contain at least two human entities, we found the most common annotations related to reasoning language is the positional description (e.g. *'on the right'*, *'behind'*). We came up with a list of commonly-used positional words like *'right', 'above'* and adjusted the keyword search rules to better filter reasoning

| | Voice | Template |
|---|---|---|
| TA | Active | the *subject predicate(active)* the *object* |
| TP | Passive | the *object* is/are *predicate(passive)* by the *subject* |
| FA | Active | the *object predicate(active)* the *subject* |
| FP | Passive | the *subject* is/are *predicate(passive)* by the *object* |

Table 5: Template for caption construction in Active-Passive Voices (AP) dataset. **TA.** True Active. **TP.** True Passive. **FA.** False Active. **FP.** False Passive. Examples: TA: the man holds the woman. TP: the woman is held by the man. FA: the woman holds the man. FP: the man is held by the woman.

| | Person | Template |
|---|---|---|
| $TP_1$ | $P_1$ | the $p_1$ [$a_{1p_1}$ and $a_{2p_1}$ ] /[who $a_{1p_1}$ $a_{2p_1}$ ] |
| $TP_2$ | $P_2$ | the $p_2$ [$a_{1p_2}$ and $a_{2p_2}$] /[who $a_{1p_2}$ $a_{2p_2}$] |
| $FP_1$ | $P_1$ | the $p_1$ [$a_{1p_1}$ and $a_{2p_2}$ /$a_{1p_2}$] /[ who $a_{1p_1}$ $a_{2p_2}$ / $a_{1p_2}$] |
| $FP_2$ | $P_2$ | the $p_2$ [$a_{2p_1}$ and $a_{1p_2}$/ $a_{2p_2}$] /[who $a_{2p_1}$ $a_{1p_2}$/ $a_{2p_2}$] |

Table 6: Template for caption construction in Coordination (CO) dataset. $TP_1$. True Person$_1$. $TP_2$. True Person$_2$. $FP_1$. False Person$_1$. $FP_2$. False Person$_2$. $a_i p_j$. Attribute$_i$ of Person$_j$ .Examples: $TP_1$: the man [wears a white shirt and holds a controller] /[who wears a white shirt holds a controller]. $TP_2$: the woman [wears a blue shirt and sits on a sofa] /[who wears a blue shirt sits on a sofa]. $FP_1$: the man [wears a white shirt and sits on a sofa] /[ who wears a white shirt sits on a sofa]. $FP_2$:the woman [holds a controller and wears a blue shirt] /[who holds a controller wears a blue shirt].

language. We also added more phrases and words to the list by checking the annotations that contain prepositions. In addition, we try to avoid numerical descriptions in our extracted information by filtering annotations that contain numbers from one to ten. This was an effective rule based on our observation on the VG annotations.

## C Sentence Generation Templates

We use sentence construction templates (Table 5 and Table 6) and the extracted information described in Section 3 to generate a set of four sentences (two true, two false) for a given image.

## D Model Finetuning Details

We fine-tune ViLBERT and LXMERT pretrained model on their entire model layers with the Image-Text Matching learning objective only. We modify the pretraining code from the VOLTA framework (available at `https://github.com/e-bug/volta/blob/main/train_concap.py`) and adapt the hyperparameter settings for finetuning provided by the code owners. We set the learning rate to 1e-5, the training batch size to 16 and the maximum training epoch to 10 after hyperparameter search-

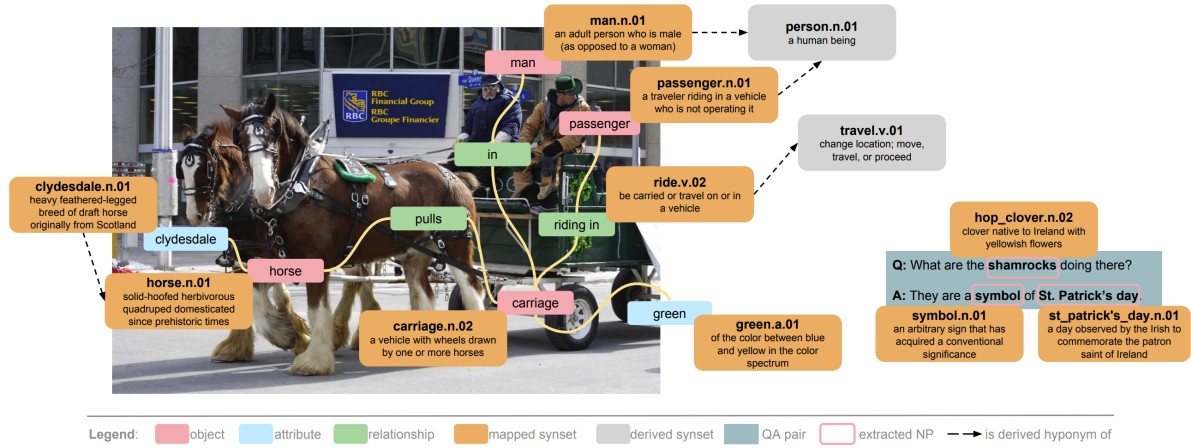

Figure 4: One example image from Visual Genome dataset with its region descriptions, QA, objects, attributes, and relationships canonicalized by Krishna et al. (2017). One example annotations for *relationships* is <predicate: pulls, subject: horse, object: carriage, ...>, for *attributes* is <carriage, green>, for *object* is <object_id, width: ..., length: ..., x:..., y:...>

ing. The model checkpoint that achieves the highest accuracy performance on the validation set will be used for evaluation. We also experiment with only fine-tuning specific layers of the models but find out that fine-tuning the whole model achieves the best performance.

## E  Human Annotation Details

We collect human annotations with Appen's internal channel option. The question interface and test question setups are shown in Figure 5.

The annotations are collected from three co-authors of this paper, including two linguistic experts and one dataset constructor. The annotators are requested to make their judgment only based on the visual and text information provided in the questions.

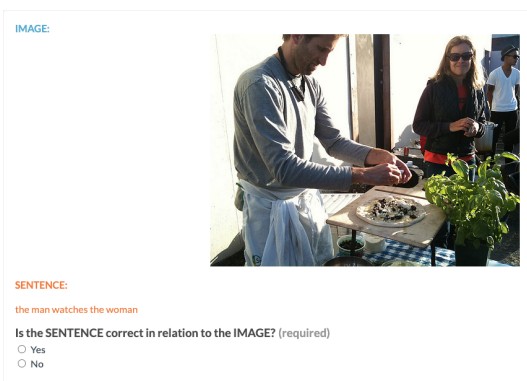

Figure 5: Example of the Appen question interface. The golden label of this question is "No".

## F  MAGMA and FROMAGe

In the zero-shot evaluation for generative models, we employ prompting to guide the models to generate constrained outputs, specifically 'yes' or 'no' responses, to perform the binary classification tasks on BLA. We use the BLIP2 prompt template and apply minor changes to it, e.g., we replace 'Question' with 'Q' and 'Answer' with 'A' following the prompting examples reported in the papers of the models. Our investigation revealed that, in over 20% of the cases, MAGMA and FRO-MAGe fail to generate the desired outputs. Indeed, many answers instead provided explanations for why a sentence is correct or incorrect. Since evaluating their performance would require more careful (including manual) analysis, in our zero-shot experiments we focused on BLIP2 and Open-Flamingo, which exhibited higher ability to adhere to the task instructions. We share the code used to preliminary test MAGMA and FROMAGe at https://github.com/shin-ee-chen/BLA.

## G  OpenFlamingo In-context Learning

After conducting preliminary in-context learning experiments, we observed that OpenFlamingo struggled to generate constrained 'yes' or 'no' answers. In particular, we experimented with two templates. The first template, similar to the BLIP2 prompt template, uses only one image input for both the examples and the question, as follows: [`<image>`Question: Is the sentence [true] appropriate for this image? yes

or no? Short Answer: yes. Question: Is the sentence [false] appropriate for this image? yes or no? Short Answer: no. Question: Is the sentence [target] appropriate for this image? yes or no? Short Answer:]. This prompt template led to nonsense outputs in most of the cases.

The second template we used closely follows the one provided in the original model paper Awadalla et al. (2023), which requires the token <image> for image input before each question, as follows: [**<image>**Question: Is the sentence [true] appropriate for this image? yes or no? Short Answer: yes.**<|endofchunk|><image>**Question: Is the sentence [false] appropriate for this image? yes or no? Short Answer: no.**<|endofchunk|><image>**Question: Is the sentence [target] appropriate for this image? yes or no? Short Answer:]. Note that the three <image> tokens always refer to the same image. With this template, the model generated more constrained answers, but only for about 40% of the cases, which is still unsatisfactory. We hypothesize this could be due, at least in part, to the properties of our questions, that are longer and more complex than the examples provided in the model paper. This could harm OpenFlamingo's ability to follow instructions.

## H   Language-Only Model on BLA Tasks

As a sanity check, we test whether the BLA tasks can be solved by a powerful text-only model, namely, GPT2 (Radford et al., 2019). We calculate the perplexity scores for the four sentences in each datapoint and rank them such that the lower the perplexity, the higher the ranking. Similarly to our experiments with CLIP, we consider two top-ranked sentences as true and the other two as false. As expected, GPT-2 performs around chance level in all tasks: 50.08% for Active-Passive, 50.47% for Coordination, and 49.96% for Relative Clause.