# OpenReview forum: "The BLA Benchmark: Investigating Basic Language Abilities of Pre-Trained Multimodal Models"
_EMNLP/2023/Conference — EMNLP 2023 Main_

### Official Review · Reviewer_5kcq · 2023-08-04

**Soundness:** 4

**Excitement:**

4: Strong: This paper deepens the understanding of some phenomenon or lowers the barriers to an existing research direction.

**Paper Topic And Main Contributions:**

This work presents a VL benchmark testing three basic linguistic constructions, namely active-passive voice, coordination, and relative clauses. The authors test in zero-shot and fine-tune settings 3 VL models on their benchmark. They find that although these are mastered by humans at an early age, VL models struggle with these tasks. Moreover, they find that the only generative models tested,  outperform the discriminative VL models both in zero-shot and fine-tuning settings, yet lagging behind human performance.

The paper is overall well-written and pleasant to read. The authors tackle an important and timely issue, which is assessing the capabilities of VL models to ground linguistic expressions which are easily mastered by humans.  This involves engaging in an evaluation that is not task-based, but linguistically motivated. The tasks chosen are fairly motivated and the benchmark construction seems solid. The main findings are fairly discussed and provide interesting insight into the grounding capabilities of pre-trained VL models, for instance, a very interesting finding is that all the models struggle with coordination even after exposure to the data. However the work focuses only on three models and three tasks, but this is discussed in the motivations and seems a reasonable improvement to be added in future work.

**Reasons To Accept:**

- new benchmark to assess linguistic capabilities of VL models
- interesting insights provided to the community

**Reasons To Reject:**

I do not identify any relevant weakness that is not already discussed in the limitation section.

**Reproducibility:**

4: Could mostly reproduce the results, but there may be some variation because of sample variance or minor variations in their interpretation of the protocol or method.

**Reviewer Confidence:**

5: Positive that my evaluation is correct. I read the paper very carefully and I am very familiar with related work.

---

> ### Author Rebuttal · Authors · 2023-08-28
>
> Thanks for your review and the positive comments!

---

### Official Review · Reviewer_pLMK · 2023-08-05

**Typos Grammar Style And Presentation Improvements:** Excellent writing.
**Soundness:** 5

**Excitement:**

4: Strong: This paper deepens the understanding of some phenomenon or lowers the barriers to an existing research direction.

**Paper Topic And Main Contributions:**

This paper tests VLMs (vision and language models) for their proficiency in linguistic constructions such as active-passive voice, coordination, and relative clauses in a new benchmark called BLA (Basic Language Abilities). The aim of this is to have a test for VL models that does not need reasoning for solving, since prior work has found that reasoning is hard enough for VL models. So the question arises of whether it is reasoning or even more basic things like linguistic understanding that they lack. The authors test 3 VL transformers in zero-shot, fine-tuning and in-context learning settings (exhaustive and complete experiments).

**Questions For The Authors:**

How do you pronounce "BLA"? Bee El A, or BLAH? 😅

**Reasons To Accept:**

The contributions of this paper are substantial:

* The paper covers a good range of VLMs (both discriminative and generative), BLIP2 is very recent and CLIP is older but powerful and popular. I highly appreciate the testing of BLIP2, a VLM with frozen linguistic representations, since I wondered whether a VLM based on a frozen language model can capture the linguistic aspects of this benchmark better. From the results, indeed it can.

* The study is well motivated and timely and I hope to see the community test even more VLMs.

* The exp 2 setting (in section 6) is extremely interesting and exhaustive (whether learning one task transfers to the other).

* The benchmark is automatically created, opening the potential for gathering more data if needed.

**Reasons To Reject:**

I don't mean to say the study is flawless, only that I cannot find flaws.

**Reproducibility:**

5: Could easily reproduce the results.

**Reviewer Confidence:**

5: Positive that my evaluation is correct. I read the paper very carefully and I am very familiar with related work.

---

> ### Author Rebuttal · Authors · 2023-08-28
>
> Thanks for your review and the positive comments! We pronounce the name of the dataset as ‘BLAH’, as it aims to test whether models comprehend or rather process language as merely blah-blah :)

---

### Official Review · Reviewer_3JNU · 2023-08-09

**Soundness:** 4

**Excitement:**

4: Strong: This paper deepens the understanding of some phenomenon or lowers the barriers to an existing research direction.

**Missing References:**

No major missing references that I am aware of.

**Paper Topic And Main Contributions:**

This paper proposes the Basic Language Abilities (BLA) benchmark, a new dataset for measuring the language abilities of vision-and-language models. The dataset is split into three types of linguistic constructions: active-passive voice, coordination, and relative clauses. A corpus of natural images and template-based sentences for each of these is constructed, with each image being paired with two correct sentences and two incorrect ones. Several vision-and-language models are benchmarked on it, and they show that existing models are significantly worse than human performance, highlighting possible areas for improvement in vision-and-language research.

**Questions For The Authors:**

- The paper shows that BLIP-2 outperforms discriminative models. It would be interesting to examine the extent to which this is due to training objectives (discriminative vs. generative), or other factors such as training data and model architecture. It would be valuable to run and include other generative model baselines, as BLIP-2 is the only generative model currently presented in the paper. For example, [OpenFlamingo](https://github.com/mlfoundations/open_flamingo), [MAGMA](https://github.com/Aleph-Alpha/magma) and [Fromage](https://github.com/kohjingyu/fromage) are some vision-and-language generative models that would be valuable to include in the evaluations.
- How do larger text-only LMs perform on this task? It would be interesting to see a text-only baseline where a model does not have access to the image. This would investigate whether it is possible to do well on BLA by simply exploiting language structure. For example, it may be interesting to see if much larger LMs (LLaMA-70B, or GPT-3) can outperform vision-and-language models.
- In Table 3, why do you think it is the case that in-context learning with BLIP-2 on BLA examples improves P_f so much, but P_t marginally? This seems to suggest that the dataset may be imbalanced or biased towards P_f somehow?
- In Table 1, does the sentence length refer to tokens or characters?

**Reasons To Accept:**

- The BLA dataset proposed in this paper seems like it will be a useful benchmark for evaluating the basic linguistic and language abilities of vision-and-language models. It appears to be a relatively difficult task, and existing SOTA vision-and-language models are significantly below human performance (and perform close to random chance).
- The paper conducts analysis on several SOTA vision-and-language models, and highlights possible directions that may be useful to explore for training better vision-and-language models (e.g., model architectures that can be prompted with in-context examples).

**Reasons To Reject:**

- **Insufficient baselines:** There is an interesting point highlighted in the paper that is not explored in enough detail. It suggests that generative models may be better than discriminative models on this task, perhaps due to their pretraining objective. However, the only generative model benchmarked in the paper is BLIP-2, which does not give us a good basis of comparison for this hypothesis. The improved results of BLIP-2 may also be due to the size of its language encoder: BLIP-2 has a much larger LM (11B) than the other models used in the paper. It will be useful to benchmark other generative vision-and-language models to ablate whether it is the size of the language encoder, pretraining objectives, or something else, that contributes to good performance on BLA.
- **Missing analysis on failure cases:** The paper would be significantly stronger if there was some analysis and visualization of the failure cases of the models. What kind of examples do the models fail on? Is this because of visual or language capabilities? A more comprehensive analysis, similar to the one in Diwan et al. (2022) would significantly strengthen the paper. In the current version of the paper, there is insufficient analysis on why the models fail exactly, and what kind of examples they tend to fail on.

**Reproducibility:**

5: Could easily reproduce the results.

**Reviewer Confidence:**

4: Quite sure. I tried to check the important points carefully. It's unlikely, though conceivable, that I missed something that should affect my ratings.

**Typos Grammar Style And Presentation Improvements:**

- The paper is generally well structured and easy to follow.
- In Appendix E, it would be helpful to list the compensation provided to the human annotators.

---

> ### Author Rebuttal · Authors · 2023-08-28
>
> Thanks for your review and comments. We address your points below:
>
> __Insufficient baselines__:  The main goal of the paper is to propose a novel dataset for evaluating the basic language abilities of current language-and-vision models and to preliminarily assess how various models representative of current ‘families’ of architectures perform on it. Yet we agree with you that testing more generative models would help to clarify whether the difference in the performance of discriminative vs. generative models we have observed generalises beyond BLIP2. To this end, we have tested Fromage (fromage_vis4 version, which uses OPT-6.7B as base LM) using the same prompts as for BLIP2. The pattern of results is very similar: like BLIP2, Fromage yields results around chance level in the zero-shot setting and improves over the discriminative models in the in-context learning setting (the in-context learning accuracy for Formage is 52.69%, 58.75% and 62.5% on AP, CO and RC, respectively). We are currently running experiments with OpenFlamingo and MAGMA (we were not able to obtain results for them yet due to time constraints). We will be happy to include the results with all these additional models to the camera-ready version of the paper. However, a systematic exploration of all available discriminative and generative models, including ablations on their pretraining objectives, parameter size, or training data, is beyond the scope of the current paper, and we leave it for future work.
>
> __Missing analysis on failure cases__: We agree that an analysis of cases where the models fail (and succeed) is interesting and can shed further light on the strengths and weaknesses of the models. Indeed, we have already carried out a preliminary qualitative analysis of a subset of each dataset, which shows interesting patterns with respect to both the type of errors that the models make (e.g., some models are shown to fail in relatively easy samples, while succeeding in more challenging samples) and the commonalities and differences between various architectures. Taking advantage of the extra page, we will include this analysis, as well as a quantitative analysis of the errors shared by various models, in the revised version of the paper. Thanks for suggesting this.
>
> __Q1__: See response to “Insufficient baselines” point above.
>
> __Q2__: Thanks for raising this point, which gives us the opportunity to further clarify why our benchmark can only be solved by a model that genuinely understands a text, a scene, and their interaction. Each sample in BLA consists of four semantically and grammatically valid sentences – we used the GRUEN language model to filter out ungrammatical/unlikely cases. Out of these linguistically valid sentences, only two apply to the target image. As such, there is no reason to expect that a language model, even if large and massively trained, should be able to perform better than chance without having access to the content of the image. We empirically tested this using GPT-2. For each sample, we obtained the model perplexity, ranked the four sentences based on it (the lower the perplexity, the higher the ranking), and computed model accuracy (see lines 409-413 in the paper about CLIP). As expected, GPT-2 was shown to perform around the chance level in all tasks: 50.08% for Active-Passive, 50.38% for Coordination, and 49.94% for Relative Clause. We will include the results of this sanity check in the updated paper.
>
> __Q3__: First, the dataset has the same number of true and false captions. False captions contain the exact same words as the correct captions while just using different word orders or switching phrases between the two sentences. Therefore, the dataset does not include any (trivial) bias toward P_f. Second, when performing in-context learning, we randomly shuffle the order for true and false examples in the context. Thus, the reported difference between P_f and P_t likely does not depend on how we use the data.
>
> Upon close inspection of the results, we conjecture that this pattern could be due to the different extent to which the model outputs a “yes” or “no” answer, e.g., 724 vs. 76 times in the RC task, respectively. In this task, the recall rate for false cases is 99.5%, while the recall for true cases is as low as 31.09%. Still, the reason why BLIP-2 outputs more positive than negative answers after in-context learning is not trivial, and surely deserves further investigation. We will add a discussion on this point in the updated version of the paper.
>
> __Q4__: In Table 1, sentence length refers to the number of tokens. We will make this clear in the paper.

---

### Meta-Review · Area_Chair_Mcng · 2023-09-18

**Recommendation:** 5

**Metareview:**

All reviewers felt very positively about this paper, viewing the proposed benchmark (evaluating VLMs on particular linguistic constructions) as useful and challenging, with interesting and insightful analysis. Once concern raised by a few reviewers is the relatively limited number of pretrained models evaluated on this dataset; however the author response adequately addressed this by including a few more (generative) models. Given that the paper is already very clearly written, it should be easy for the authors to present these extra results and the promised additional qualitative analysis in the extra page in any camera-ready version to make the paper even stronger.

---

### Decision · Program_Chairs · 2023-10-07

**Decision:**

Accept-Main

**Comment:**

All reviewers felt very positively about this paper, viewing the proposed benchmark (evaluating VLMs on particular linguistic constructions) as useful and challenging, with interesting and insightful analysis. Once concern raised by a few reviewers is the relatively limited number of pretrained models evaluated on this dataset; however the author response adequately addressed this by including a few more (generative) models. Given that the paper is already very clearly written, it should be easy for the authors to present these extra results and the promised additional qualitative analysis in the extra page in any camera-ready version to make the paper even stronger.